# StringLLM: Understanding the String Processing Capability of Large Language Models

**Xilong Wang**[1,2]**, Hao Fu**[2]**, Jindong Wang**[3]**, Neil Zhenqiang Gong**[1]
[1]Duke University, [2]Li Auto, [3]William & Mary
{xilong.wang, neil.gong}@duke.edu, fuhao8@lixiang.com
jwang80@wm.edu

## Abstract

String processing, which mainly involves the analysis and manipulation of strings, is a fundamental component of modern computing. Despite the significant advancements of large language models (LLMs) in various natural language processing (NLP) tasks, their capability in string processing remains underexplored and underdeveloped. To bridge this gap, we present a comprehensive study of LLMs' string processing capability. In particular, we first propose *StringLLM*, a method to construct datasets for benchmarking string processing capability of LLMs. We use StringLLM to build a series of datasets, referred to as *StringBench*. It encompasses a wide range of string processing tasks, allowing us to systematically evaluate LLMs' performance in this area. Our evaluations indicate that LLMs struggle with accurately processing strings compared to humans. To uncover the underlying reasons for this limitation, we conduct an in-depth analysis and subsequently propose an effective approach that significantly enhances LLMs' string processing capability via fine-tuning. This work provides a foundation for future research to understand LLMs' string processing capability. Our code and data are available at https://github.com/wxl-lxw/StringLLM.

## 1 Introduction

String processing is one of the most essential tasks in modern computing. It is mainly involved with string analysis and manipulation such as accessing characters at a specific index (string indexing) or locating a substring within a string (substring searching). For instance, search engines match user queries to relevant documents using substring searching, with Google processing over 3.5 billion searches per day. String processing is also crucial for searching through large databases, such as Amazon's over 400 million product listings. It is also vital for cleaning and normalizing messy text data in machine learning. Additionally, string processing plays a critical role in LLM reasoning. For instance, the chain-of-thought examples in the OpenAI o1 documentation (OpenAI, 2024b), such as cipher and crossword problems, frequently involve string processing. Programming languages like Python offer powerful built-in functions for string manipulation, making it easier for *humans* to solve complex string-related problems.

With the advance of LLMs on natural language processing (NLP) tasks (OpenAI, 2024b;a; 2023a), it is intuitive to assume that LLMs should also achieve great performance on string-related problems since strings are texts. However, the reality is that LLMs often struggle with these seemingly simple challenges. A popular example (Goodside, 2024) shows that GPT-4o (OpenAI, 2024a) frequently fails in counting the number of "r"s in the word "strawberry". Additionally, the use of LLMs for string processing remains largely unstudied. Existing works (Shin & Kaneko, 2024; Tan et al., 2024; Yehudai et al., 2024; Zhou et al., 2023) are mostly limited to case studies with only a few string processing tasks, lacking a comprehensive evaluation of LLMs' capabilities and limitations in this domain. Consequently, it remains unclear how well LLMs can handle such tasks, why they fail, and what might improve their performance.

---

Work done when Xilong Wang was an intern at Li Auto mentored by Hao Fu.

**StringLLM:** In this paper, we conduct the first comprehensive study toward understanding the string processing capability of LLMs. One core challenge is how to create large representative datasets with a diverse set of string processing tasks, different types of strings, and guaranteed ground truth answers. To address the challenge, we propose *StringLLM*, the *first* method to construct datasets for benchmarking string processing capability of LLMs. StringLLM begins by manually collecting a set of fundamental string processing tasks called *atomic tasks*. Then, StringLLM combines these atomic tasks to create more intricate ones, referred to as *composite tasks*. Given a task, StringLLM generates question templates for it, which are then used to derive question inputs. Finally, StringLLM obtains the ground truth answer for a question input by executing the Python code of the corresponding task, generating question-answer pairs for string processing.

**Crafting new datasets:** We use StringLLM to construct a series of datasets, referred to as *String-Bench*. These datasets include 1,511 string processing tasks spanning a broad range of applications, and three common types of strings: hash strings, multilingual natural language strings, and random strings. StringBench allows us to thoroughly assess the ability of LLMs to process strings across various scenarios.

**Benchmarking string processing capabilities of LLMs:** Building on StringBench, we present the *first* systematic evaluation of LLMs on string processing tasks, using three prompting strategies: raw instructions, Chain of Thought (CoT) (Wei et al., 2022), and Program of Thought (PoT) (Chen et al., 2022). Our comprehensive experimental results demonstrate that: 1) LLMs struggle with string processing compared to human capability. In particular, they achieve a maximum of 48.89% accuracy using raw instructions; 2) LLMs' performance varies across datasets, revealing significant disparities in their ability to process different types of strings. Specifically, random strings are the most challenging, with accuracy peaking at 43.94% using raw instructions; and 3) Prompt engineering significantly improves performance. Some LLMs show over a 20% improvement when using PoT compared to raw instructions, offering valuable insights for designing better solutions.

**Understanding why LLMs struggle with string processing:** We conduct the *first* in-depth analysis to investigate why LLMs struggle with string processing. Our analysis reveals that: 1) Tokenization fails to split strings into individual characters, resulting in a lack of character-level understanding in LLMs; and 2) Token embedding lacks character-level information such as token length information, further highlighting LLMs' limited character-level comprehension of strings. Our analysis and Yehudai et al. (2024) show that Transformers have limited ability in solving string processing tasks. To address this, we propose an effective solution to enhance LLMs' performance in string processing, without altering the architecture of Transformers. Utilizing our well-constructed StringBench, we conduct supervised fine-tuning on three different open-source LLMs. Our fine-tuned models improve average test accuracy of our datasets by at least **38.80%**, compared to the best-performing prompt engineering technique, PoT. We then evaluate the foundational capabilities of our fine-tuned models on three general-purpose benchmarks. The results show that the string processing capabilities of our fine-tuned models are significantly enhanced without substantially degrading their foundational capabilities. Specifically, the three fine-tuned LLMs sacrifice at most 1.35% on average across the three general-purpose benchmarks.

## 2 RELATED WORK

**LLMs' string processing capability is underexplored:** LLMs have advanced significantly in recent years, demonstrating impressive capabilities across diverse NLP tasks, such as reasoning (Wei et al., 2022; Chen et al., 2022), coding (Roziere et al., 2023; Zhu et al., 2024), and instruction following (Ouyang et al., 2022). However, their string processing capabilities remain understudied. Shin & Kaneko (2024) explored LLMs' ability to handle character-level tasks as opposed to token-level tasks. Their study revealed that LLMs struggle with simple character-level operations (e.g., character insertion, deletion, reordering), which humans can perform with ease. In contrast, LLMs' performance on token-level tasks (e.g., sentence retrieval, word insertion) is relatively stronger, highlighting a discrepancy between token and character-level understanding. Zhou et al. (2023) proposed a benchmark to evaluate the instruction-following capabilities of LLMs, including some string processing tasks. However, the tasks are few and lack comprehensiveness, limiting their use to case studies. Tan et al. (2024) further noted that LLMs often struggle with basic counting tasks,

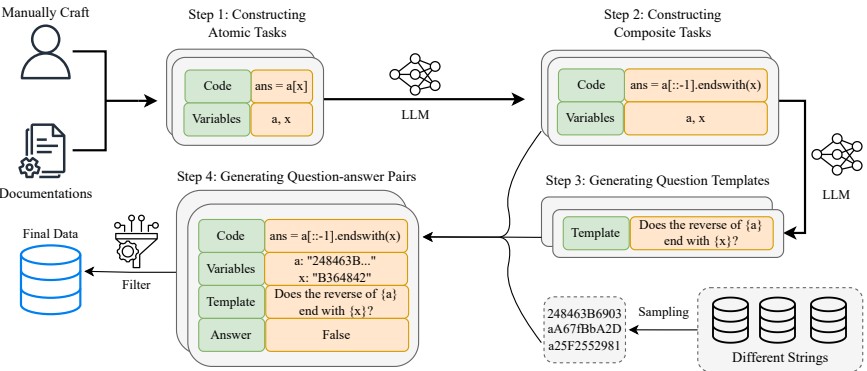

Figure 1: Overview of how StringLLM builds the benchmark datasets. *Code* represents Python code expression for each string processing task, *Template* denotes the generated question template for each task, *Variables* are the placeholders within both *Code* and *Template*, and *Answer* refers to the groundtruth for each sample.

such as generating a paragraph with a specific word count and then correctly identifying that number. Correspondingly, Yehudai et al. (2024) discussed strategies to address this counting problem in Transformers. However, these works are limited to case studies focused on single or a few string processing tasks, lacking a comprehensive evaluation. Furthermore, they do not provide an in-depth analysis of why LLMs struggle with accurately processing strings, nor do they offer solid experimental evidence to support their conclusions. Consequently, they fail to propose effective solutions to enhance string processing capability of LLMs.

# 3 CONSTRUCTING DATASETS VIA STRINGLLM

## 3.1 OVERVIEW

An overview of StringLLM is shown in Figure 1. We begin by creating tasks. Specifically, we first create tasks that are fundamental and cannot be broken down further, referred to as *atomic tasks*. These are either manually designed by us or sourced from public programming language documentation. Once we have these atomic tasks, we combine multiple of them into more complex tasks, referred to as *composite tasks*. Using both atomic and composite tasks, we generate natural language question-answer pairs for LLMs, resulting in a series of datasets called *StringBench*. It includes three distinct datasets, each designed to query LLMs in different ways:

- **Multilingual**: Queries LLMs to process multilingual natural language strings.
- **Hash**: Queries LLMs to process strings encoded by different hash functions.
- **Random Strings**: Queries LLMs to process random strings composed of printable characters.

The three datasets cover diverse usage of LLMs, including natural language tasks and code-related tasks. The Multilingual dataset focuses on natural language tasks with special attention to multiple languages. Hash strings represent a typical type of strings in code-related tasks. To cover a wider range of characters, we also consider random strings of all printable characters. Manually constructing datasets of this nature is extremely time-consuming and impractical. We leverage GPT-4o (OpenAI, 2024a) to assist in the data construction process. We avoid direct use of GPT-4o for string processing, as we will show in Section 4 that it struggles with string-related tasks. Instead, we make use of its coding abilities, including code generation, understanding, and summarization. We take code as a proxy to produce accurate datasets. Previous study (OpenAI, 2024a) has shown that GPT-4o performs exceptionally well in these areas, achieving a score of 90.2% on the HumanEval benchmark (Chen et al., 2021). This highlights its remarkable capabilities in code-related applications, enabling us to use it for data construction.

## 3.2 Constructing String Processing Tasks

**Atomic string processing tasks:** We begin by thoroughly examining the built-in function documentation [1] of Python to identify atomic string processing tasks. Specifically, we focus on distinguishing unique functions while filtering out those that are redundant or exhibit overlapping functionalities. For example, the `splitlines()` function, which is designed for splitting strings at line boundaries, is obviously a specialized case of the more general `split()` function. Therefore, we integrate `splitlines()` into the broader `split()` task and exclude it from separate consideration. Following this, a set of atomic string processing tasks is obtained. However, this initial set proves to be insufficient for our comprehensive needs. Many common tasks are not encapsulated within the Python built-in functions. To address this gap, we manually develop an additional set of atomic tasks. These include fundamental tasks such as indexing, slicing, and concatenation. They are frequently used but are not explicitly provided as built-in functions. Once we have determined the complete set of atomic tasks, we proceed to formally express each task in Python code. We manually create Python scripts that perform these tasks using placeholder variables. These scripts then serve as templates for constructing composite tasks (see Table 12 in Appendix for examples).

By manually reviewing the built-in function documentation and crafting an additional set of atomic tasks, we ensure that these atomic tasks are representative and can compose many diverse composite tasks. Moreover, by filtering out tasks that are redundant or exhibit overlapping functionalities, we avoid the selected atomic tasks having conflicted meanings that could form bad composite tasks.

**Composite string processing tasks:** Building on atomic tasks, we iteratively generate more complex string processing tasks. In each iteration, we randomly shuffle and then input all atomic tasks into GPT-4o, combining them into composite tasks. Specifically, we prompt GPT-4o to generate Python code for composite tasks based on the code of atomic tasks. This approach fully leverages GPT-4o's Program of Thought (PoT) capability, instead of using it for direct string processing. As a result, we generate a total of 1,462 composite tasks, covering a wide range of complicated string processing scenarios commonly used in practice.

## 3.3 Generating Question-answer Pairs for Each Task

**Constructing question templates:** After obtaining the atomic and composite tasks, we once again use GPT-4o to generate question templates for these tasks. As illustrated in Figure 1, a question template describes the task with placeholders. Given the Python code, GPT-4o is prompted to create a template that asks how the task described by the code would be accomplished. This ensures that the question templates are aligned with the tasks defined by the Python code. Thus, such templates can effectively guide LLMs to perform the required tasks. This procedure leverages GPT-4o's capability of code summarization, without requiring it to process strings directly. As with previous steps, this approach skillfully

Table 1: Dataset statistics.

| Dataset | Atomic Tasks | Composite Tasks | Question-answer Pairs |
|---|---|---|---|
| Multilingual | 49 | 1,462 | 22,161 |
| Hash | 49 | 1,462 | 18,441 |
| Random String | 49 | 1,462 | 21,159 |

exploits GPT-4's strengths while minimizing its limitations. To promote diversity, we generate three paraphrased question templates for each task. Specifically, to ensure the quality of the templates, we first instruct GPT-4o to generate 10 templates. Then, we have GPT-4o itself rank these generated templates and select the top three.

**Generating question-answer pairs based on the template:** Once the question templates are generated, we replace the placeholders in the templates with specific strings to be processed. This procedure transforms the templates into question inputs. To promote the diversity of our dataset, we create three types of strings to each question template. This results in StringBench, containing three distinct datasets:

---

[1] https://docs.python.org/3/library/stdtypes.html#string-methods

- **Multilingual:** We randomly sample strings from the Flores-200 dataset (Costa-jussà et al., 2022), which is a machine translation dataset including data in 200 languages.
- **Hash:** We first randomly sample strings from the Flores-200 dataset. For each string, we randomly select a hash function from a set of 10 different functions to encode it. The 10 used cryptographic hash functions are listed in Table 11 in Appendix.
- **Random String:** We randomly sample strings using all printable characters in Python, including letters, numbers, punctuation, escape characters, and other special symbols.

Finally, we execute the Python code to generate the ground truth answers for all datasets. This ensures a reliable and accurate reference for both training and evaluation purposes. The guaranteed ground truth data is a significant strength of our approach. Unlike other datasets that may suffer from inconsistencies or ambiguities in labeling, our ground truth provides a definitive reference, improving the reliability of evaluations. This rigorous standard allows us to measure LLMs' performance with high precision and offers a robust foundation for training LLMs to achieve better performance. By leveraging this guaranteed ground truth, we ensure that our datasets serve as a valuable benchmark for both research and practical applications. Statistics of our datasets are shown in Table 1, and randomly selected question-answer pairs from each dataset are presented in Table 2.

Table 2: Randomly selected question-answer pairs from our datasets.

| Dataset | Question | Answer |
|---------|----------|--------|
| Multilingual | Retrieve the first 29 characters from string `"Felicia, rakamboita dutu rechikamu chechina pa Sikero yeSaffir-Simpson Hurricane, yakaneta kusvika kuderera kwetropical ichizopra neChipiri"`. | `"Felicia, rakamboita dutu rech"` |
| Hash | Get the index of the first instance of `"dc"` in string `"c11c8a595476dcde4f91a8 dce2acaba2"`. | 12 |
| Random String | Take every 4th character from the start of string `"N/5qe!wj8U*8dvsN/am'UGfN /A=n+%$5)3HA?d#Jn&F4&,(WG-p:1Vw]"` up to the 50th character. | `"Ne8d/U/ +)?n&G"` |

## 3.4 POST-PROCESSING

We filter out duplicate and low-quality data after the above steps. Duplicate code and question templates are removed in Steps 2 and 3 in Figure 1. All Python code is validated through execution, and any code that results in execution errors is also removed.

## 4 BENCHMARKING LLMS' STRING PROCESSING CAPABILITIES

### 4.1 EXPERIMENTAL SETUP

**LLMs:** We use the following LLMs for our evaluation: GPT-4-Turbo (OpenAI, 2023b), GPT-4o (OpenAI, 2024a), GPT-3.5 (OpenAI, 2022), DeepSeek-Chat (DeepSeek-AI, 2024), Gemma-2-9b (Google, 2024), Llama-3.1-8B (Meta, 2024), Mistral-7B-v0.3 (Mistral-AI, 2024c), Mathstral-7B-v0.1 (Mistral-AI, 2024b), Codestral-22B-v0.1 (Mistral-AI, 2024a) and DeepSeek-Coder (Zhu et al., 2024). Table 10 in Appendix shows more details of the models.

**Training-test split of our datasets:** For the test sets, we randomly split 20% of the data from each of the three datasets—Multilingual, Hash, and Random String. We ensure that the test sets cover the full range of 1,511 string processing tasks in our datasets. This approach guarantees that the diversity and complexity of these tasks are well-represented in the test sets, allowing a comprehensive evaluation of the model's performance. The remaining 80% of our datasets is used as the training sets for our experiments on fine-tuning LLMs in Section 6.

**Prompt engineering:** In our experiment, we apply three prompt engineering techniques to evaluate the performance of LLMs. 1) **Raw instructions:** The questions are input directly to the LLMs without any additional guidance, allowing LLMs to choose their own methods for processing the strings. 2) **Chain of Thought (CoT)** (Wei et al., 2022): Prompts LLMs to generate step-by-step solutions for string processing tasks. By breaking down problems into smaller steps, this technique helps LLMs handle complex reasoning tasks more systematically. 3) **Program of Thought (PoT)**

Table 3: Accuracy for string processing capability of humans and different LLMs.

| LLM | Method | Multilingual | Hash | Random String | AVG |
|---|---|---|---|---|---|
| GPT-4o | Raw Inst. | 43.09 | 48.89 | 43.94 | 45.31 |
| | CoT | 50.05 | 52.01 | 45.63 | 49.23 |
| | PoT | 69.19 | 68.00 | 49.06 | 62.08 |
| GPT-4-Turbo | Raw Inst. | 35.49 | 41.61 | 33.87 | 36.99 |
| | CoT | 43.98 | 48.10 | 43.09 | 45.06 |
| | PoT | 66.50 | 71.01 | 48.74 | 62.08 |
| GPT-3.5 | Raw Inst. | 8.36 | 15.93 | 13.60 | 12.63 |
| | CoT | 30.81 | 29.94 | 24.77 | 28.51 |
| | PoT | 42.63 | 42.62 | 25.26 | 36.84 |
| DeepSeek-Coder | Raw Inst. | 17.79 | 21.34 | 13.97 | 17.70 |
| | CoT | 22.87 | 29.72 | 22.87 | 25.15 |
| | PoT | 54.85 | 57.46 | 29.42 | 47.24 |
| DeepSeek-Chat | Raw Inst. | 4.26 | 3.50 | 1.90 | 3.22 |
| | CoT | 7.89 | 9.99 | 7.30 | 8.39 |
| | PoT | 7.12 | 7.76 | 2.39 | 5.76 |
| Gemma-2-9b | Raw Inst. | 21.49 | 21.01 | 14.72 | 19.07 |
| | CoT | 23.81 | 24.67 | 18.84 | 22.44 |
| | PoT | 55.34 | 52.71 | 14.49 | 40.85 |
| Mistral-7B-v0.3 | Raw Inst. | 4.54 | 3.83 | 2.67 | 3.68 |
| | CoT | 10.56 | 8.77 | 6.02 | 8.45 |
| | PoT | 32.54 | 31.57 | 14.07 | 26.06 |
| Mathstral-7B-v0.1 | Raw Inst. | 4.85 | 5.81 | 2.08 | 4.25 |
| | CoT | 13.25 | 15.72 | 10.07 | 13.01 |
| | PoT | 21.40 | 25.95 | 12.31 | 19.89 |
| Codestral-22B-v0.1 | Raw Inst. | 19.03 | 15.99 | 12.52 | 15.85 |
| | CoT | 16.08 | 24.46 | 10.56 | 17.03 |
| | PoT | 23.92 | 14.39 | 15.75 | 18.02 |
| Llama-3.1-8B | Raw Inst. | 11.87 | 16.61 | 10.30 | 12.93 |
| | CoT | 19.85 | 22.04 | 17.13 | 19.67 |
| | PoT | 39.35 | 42.62 | 21.79 | 34.59 |
| Human | Manual | 42.27 | 95.74 | 93.55 | 77.19 |
| | Python | 98.12 | 97.99 | 98.61 | 98.24 |

(Chen et al., 2022): Prompts LLMs to generate responses mainly in programming language. This technique structures the reasoning process of LLMs in a programmatic manner. Thus, it enables LLMs to perform intricate string processing tasks with greater precision.

**Evaluation metrics:** To evaluate the performance of LLMs on string processing tasks, we use *Accuracy (Acc)*. It measures the correctness of LLMs' responses to input questions. As mentioned in Section 3, we execute the Python code of each task to obtain the ground truth answer. Hence, due to the deterministic nature of these answers, correctness is determined by exact matches between the model's final output and the ground truth answer.

**Human study:** We conducted a human study to evaluate the string processing capabilities of LLMs in comparison with humans. We ask human annotators to perform all string processing tasks in the test set, either manually or by writing Python code.

## 4.2 EXPERIMENTAL RESULTS

**LLMs struggle to process strings compared to humans:** The experimental results in Table 3 reveal that all LLMs struggle with string processing tasks across all datasets and prompt engineering techniques. For example, when using raw instruction prompts, even the best-performing LLMs, GPT-4o and GPT-4-Turbo, achieve only 36.99% and 45.31% Acc on average, respectively. Other LLMs perform significantly worse. For instance, GPT-3.5 reaches an average Acc of just 12.63%, which is 32.68% lower than its updated version GPT-4o. LLMs with much fewer parameters exhibit

even more significant performance drops, with average Acc below 20%. Specifically, Mistral-7B-v0.3 and DeepSeek-Chat exhibit particularly poor performance. They achieve average Acc of only 3.68% and 3.22%, respectively. In contrast, humans demonstrate near-perfect performance across all datasets. They do struggle to process strings manually, as they cannot identify and segment Multilingual characters. For instance, in the case of the Arabic language, non-native speakers may find it challenging to process such strings due to the difficulty of identifying individual characters. However, when using Python code to process strings, humans achieve an average Acc of 98.24%. This highlights the limitations of LLMs compared to human capability.

**Random strings are harder to process:** As shown in Table 3, LLMs exhibit significant performance variation across our datasets. In general, LLMs perform better on the Hash dataset compared to the Random String and Multilingual datasets. For instance, with raw instruction prompts, GPT-4o achieves 48.89% Acc on the Hash dataset. This is 5.8% higher than its performance on the Multilingual dataset and 4.95% higher than on the Random String dataset. Overall, 6 out of 10 LLMs perform best on the Hash dataset, indicating that LLMs excel in processing hash strings. Additionally, 8 out of 10 LLMs show their worst performance on the Random String dataset when using raw instruction prompts. This finding suggests that LLMs lack expertise in processing random strings. One possible reason is that LLMs are not trained on data containing random strings, making it an out-of-distribution problem.

**Prompt engineering improves performance:** Table 3 shows that LLMs perform largely better with PoT and CoT than with raw instructions, where all 10 LLMs exhibit their lowest average Acc. In contrast, when prompted with PoT, LLMs consistently deliver the best results. Specifically, 9 out of 10 LLMs show their highest performance. For example, the DeepSeek-Coder achieves an average Acc of 47.24% across the three datasets when using PoT. It is 22.09% and 29.54% higher than with CoT and raw instructions, respectively. This is likely because, compared to CoT, PoT structures the reasoning process more hierarchically and incorporates programming to solve problems. As a result, it is better suited for precise character-level manipulation required in string processing. Examples of GPT-4o solving a string processing task using different prompt engineering techniques are shown in Figure 2 - 3 in Appendix. These examples further highlight PoT's advantage over CoT and raw instructions in string processing.

## 5 UNDERSTANDING WHY LLMS STRUGGLE WITH STRING PROCESSING

### 5.1 TOKENIZATION CANNOT SPLIT STRINGS INTO CHARACTERS

Tokenization is the process of breaking down texts into smaller units called tokens, which can be words, subwords, or characters. This process is fundamental to how LLMs process and understand text. However, LLMs lose character-level details of input, when it is tokenized. When an input text is tokenized into subwords or words, LLMs may lose the granular character-level details of the input. This can lead to issues for string processing tasks, which require precise character-level manipulation or analysis. For instance, checking if a string is a palindrome (reads the same backward as forward) can be problematic due to tokenization. Taking a simple example, the phrase `"A man a plan a canal Panama"` should be recognized as a palindrome when ignoring spaces and case differences. However, the tokenizer might split it into `["A", "man", "a", "plan", "a", "canal", "Panama"]`. As a result, LLMs may fail to perform the check correctly, since they may treat each word as a separate token instead of considering the entire character sequence.

Table 4 shows ratios of actual length to the number of tokens for sampled strings in our datasets, when tokenized by different LLMs. From Table 4, we can draw the following conclusion:

- For the Multilingual dataset, as shown in Table 4, all three LLMs show the highest ratio. This is because in multilingual natural language strings, words and subwords remain indivisible during tokenization. Consequently, tokenization does not break down the strings into finer fragments. This could contribute to lower Acc in such scenarios.
- For the Hash dataset, tokenization breaks the string into finer fragments. As shown in Table 4, two out of three LLMs display the lowest ratio. Furthermore, hash strings typically consist of alphanumeric characters without spaces or other natural language patterns. They also have a fixed length, making them relatively structured. Additionally, they generally do not contain punctuation

or escape characters, further simplifying their structure. In this context, hash strings are easier for LLMs to process and analyze, leading to higher Acc.

- For the Random String dataset, tokenization segments the strings into smaller units as well. However, these strings often contain escape characters, punctuation, and other special characters. These elements add complexity to the structure of random strings, making it challenging for LLMs to analyze and understand them accurately.

Correspondingly, as shown in Table 3, LLMs perform best on the Hash dataset due to the simpler, more structured nature of the strings. For the Multilingual dataset, performance is lower because tokenization preserves the integrity of words and subwords. Consequently, it could be more difficult for LLMs to establish character-level understanding of such strings. For the Random String dataset, the added complexity from special characters further challenges LLMs, resulting in the lowest Acc.

Table 4: Ratios of actual length to the number of tokens for sampled strings in our datasets, when tokenized by different LLMs.

| LLM | Multilingual | Hash | Random String |
|---|---|---|---|
| Gemma-2-9b | 2.53 | 1.12 | 1.30 |
| Mistral-7B-v0.3 | 1.73 | 1.06 | 1.15 |
| Llama-3.1-8B | 2.13 | 1.67 | 1.32 |

## 5.2 TOKENIZATION MAKES LLMs LACK CHARACTER-LEVEL UNDERSTANDING

As stated in the previous section, tokenization typically splits strings into word-level or subword-level tokens instead of individual characters. However, existing studies do not provide sufficient evaluation to confirm that these tokens lack character-level information. In this section, we aim to further demonstrate that current tokenization, which cannot perform such fine-grained splitting, makes LLMs indeed lack character-level information. Table 5 shows the string processing capability of different LLMs, where white spaces are inserted between all characters of strings. This insertion of white spaces aims to segment every character in strings, avoiding the use of word-level or subword-level tokens, thereby introducing potential character-level information to LLMs.

Table 5: Accuracy for string processing capability of different LLMs, where white spaces are inserted between all characters of strings.

| LLM | Method | Multilingual | Hash | Random String | AVG |
|---|---|---|---|---|---|
| Gemma-2-9b | Raw Inst. | 20.48 | 21.23 | 14.31 | 18.67 |
| | CoT | 24.22 | 25.16 | 20.07 | 23.15 |
| | PoT | 55.69 | 52.73 | 18.98 | 42.47 |
| Mistral-7B-v0.3 | Raw Inst. | 4.97 | 3.99 | 2.79 | 3.92 |
| | CoT | 12.45 | 10.53 | 7.34 | 10.11 |
| | PoT | 32.28 | 32.67 | 15.32 | 26.76 |
| Llama-3.1-8B | Raw Inst. | 12.83 | 16.56 | 9.76 | 13.05 |
| | CoT | 20.64 | 23.21 | 19.20 | 21.02 |
| | PoT | 42.01 | 43.30 | 24.74 | 36.68 |

Compared to Table 3, the performance of LLMs is improved, indicating that the word-level or subword-level tokens make LLMs lack character-level understanding. In contrast, augmenting the tokenizer with finer-grained character-level segmentation could be a promising direction for improving LLM performance.

## 5.3 LLMs' UNDERSTANDING OF STRINGS IS LIMITED

Our experimental results have demonstrated that tokenization cannot split strings into individual characters, and word-level or subword-level tokens do not include sufficient character-level information. Considering the architecture of Transformers, tokenization is the starting point, and its limitations can lead to further errors. Consequently, the attention mechanism and entire Transformer architecture of LLMs cannot effectively analyze character-level information, leading to a lack of fundamental understanding of strings.

Table 6: Acc for string processing capability of different LLMs before and after fine-tuning, when prompted with PoT.

| LLM | | Multilingual | Hash | Random String | Avg. Change |
|---|---|---|---|---|---|
| Gemma-2-9b | Before | 55.34 | 52.71 | 14.49 | + 38.80 |
| | After | 82.49 | 87.98 | 68.46 | |
| Mistral-7B-v0.3 | Before | 32.54 | 31.57 | 14.07 | + 56.51 |
| | After | 83.24 | 90.28 | 74.18 | |
| Llama-3.1-8B | Before | 39.35 | 42.62 | 21.79 | + 42.87 |
| | After | 79.83 | 85.64 | 66.91 | |

Table 7: Acc/Acc-Norm for foundational capabilities of different LLMs before and after fine-tuning.

| LLM | | MMLU | Hellaswag | ARC | Avg. Change |
|---|---|---|---|---|---|
| Gemma-2-9b | Before | 71.88 | 80.08 | 64.76 | - 0.87 |
| | After | 70.92 | 80.39 | 62.80 | |
| Mistral-7B-v0.3 | Before | 59.74 | 82.90 | 58.62 | - 1.35 |
| | After | 57.56 | 82.14 | 57.51 | |
| Llama-3.1-8B | Before | 67.94 | 79.33 | 55.03 | + 0.51 |
| | After | 66.24 | 79.48 | 58.11 | |

## 6 IMPROVING STRING PROCESSING CAPABILITY VIA FINE-TUNING

In this section, we further explore the efficacy of fine-tuning in improving the performance of LLMs. We use the training sets described in Section 4.1 to fine-tune three LLMs: Llama-3.1-8B (Meta, 2024), Gemma-2-9b (Google, 2024), and Mistral-7B-v0.3 (Mistral-AI, 2024c), with the help of the LlamaFactory framework (Zheng et al., 2024) and LoRA (Hu et al., 2022). As shown in Table 3, PoT performs the best across our three datasets compared to raw instructions and CoT. Therefore, we structure our questions to guide the LLMs in generating PoT solutions, using Python code from our datasets as expected output. Additionally, we include general-purpose datasets (Alpaca-GPT-4 (Peng et al., 2023) and Dolly-15k (Conover et al., 2023)) and programming datasets (Code Alpaca (Chaudhary, 2023) and OpenCoder (Huang et al., 2024)) to improve the robustness of fine-tuned models. The inclusion of general-purpose datasets in fine-tuning is inspired by state-of-the-art coding LLMs (Roziere et al., 2023; Zhu et al., 2024), which use general-purpose datasets during their training phases. The inclusion of such datasets is a common practice to ensure the models have a certain level of foundational capabilities. Hence, excluding general datasets would compromise the foundational capabilities of the models, which is shown in Table 9. We continue to use the Acc as the primary evaluation metric for our datasets. Additionally, to evaluate the foundational capabilities of the LLMs, we utilize three datasets: MMLU (Hendrycks et al., 2021), Hellaswag (Zellers et al., 2019), and ARC (AI2 Reasoning Challenge) (Clark et al., 2018). We evaluate LLMs on these benchmarks using the LM-Evaluation-Harness framework (Gao et al., 2024), and all evaluations are conducted in a zero-shot setting. We use **Acc** or **Acc-Norm** stated in the framework as an evaluation metric. Specifically, for Hellaswag and ARC, we use **Acc-Norm**; for MMLU, we use **Acc**.

Table 6 shows the string processing capability of LLMs when prompted with PoT, while Table 7 shows the foundational capabilities of LLMs and Table 8 shows their performance on code generation benchmarks, both before and after fine-tuning. From these results, we can draw the following conclusions: **1) Our fine-tuning is effective.** The results in Table 6 clearly show that the performance of all three LLMs significantly improves after fine-tuning. For instance, the Mistral-7B-v0.3 (Mistral-AI, 2024c) model's Acc on the Random String dataset improves by **60.11%**. This finding demonstrates that our fine-tuning effectively enhances the LLMs' capability in string processing. **2) The fine-tuned LLMs maintain their foundational capabilities.** Table 7 demonstrates that our fine-tuning has minimal impact on the foundational capabilities of LLMs. The results show that all evaluated LLMs exhibit an average performance degradation of less than **1.35%**. In some cases, performance even improves, likely due to the inclusion of two general-purpose datasets during fine-tuning. This finding highlights that our fine-tuning not only enhances LLMs' string processing

Table 8: Performance of various LLMs on HumanEval, HumanEval+, MBPP, and MBPP+ benchmarks before and after fine-tuning.

| LLM | | HumanEval | HumanEval+ | MBPP | MBPP+ | Avg. Change |
|---|---|---|---|---|---|---|
| Gemma-2-9b | Before | 67.7 | 59.1 | 73.3 | 63.0 | - 0.78 |
| | After | 66.5 | 59.1 | 72.8 | 62.2 | |
| Mistral-7B-v0.3 | Before | 36.0 | 31.1 | 50.3 | 42.1 | - 0.68 |
| | After | 34.8 | 30.5 | 49.2 | 42.3 | |
| Llama-3.1-8B | Before | 64.5 | 55.5 | 68.0 | 55.6 | - 1.05 |
| | After | 63.4 | 54.3 | 67.5 | 54.2 | |

Table 9: Acc/Acc-Norm for foundational capabilities of different LLMs before and after fine-tuning, where LLMs are finetuned without general-purpose datasets.

| LLM | | MMLU | Hellaswag | ARC | Avg. Change |
|---|---|---|---|---|---|
| Gemma-2-9b | Before | 71.88 | 80.08 | 64.76 | - 1.49 |
| | After | 71.41 | 79.76 | 61.09 | |
| Mistral-7B-v0.3 | Before | 59.74 | 82.90 | 58.62 | - 1.60 |
| | After | 57.72 | 82.52 | 56.23 | |
| Llama-3.1-8B | Before | 67.94 | 79.33 | 55.03 | - 0.46 |
| | After | 65.65 | 80.60 | 54.68 | |

ability but also preserves their foundational capabilities. **3) The fine-tuned LLMs maintain their coding capabilities.** Results in Table 8 indicate that the code generation performance of LLMs shows a slight decline (no more than **1.05%** on average) before and after fine-tuning. However, our fine-tuning method is quite naive, since it is not our primary focus. AI companies with advanced model training pipelines and substantial computational resources can better integrate our dataset with general and code-specific data, enabling them to maintain both the general and coding capabilities of LLMs.

Comparing finetuning with alternative approaches operating at the tokenizer level is another interesting direction. As a brief investigation, Table 13 in Appendix shows LLMs' performance where the characters in the string are encoded separately. In this setup, each character is encoded by its token ID in the vocabulary, rather than being processed as part of a complete word or subword. Surprisingly, the performance of LLMs degrades compared to Table 3 in the paper. One possible explanation is that this essentially employs a different tokenizer, as it encodes the same input text in a different way. The performance drops since LLMs are trained on the original tokenizer. Retraining LLMs with new finer-grained tokenizers would be another alternative approach. Exploring such approaches would be an exciting avenue for future LLM developers, but it goes beyond the scope of our work. Additionally, retraining an LLM with a new tokenizer would require significant computational resources, which are unavailable to us.

## 7 CONCLUSION

In this paper, we presented the *first* comprehensive study on LLMs' capability in string processing. we proposed *StringLLM*, a method to construct datasets for benchmarking string processing capability of LLMs. Following this, we used StringLLM to develop a series of datasets covering a broad range of string processing tasks and different types of strings. Extensive experiments conducted on these datasets indicated that current LLMs have limited capability in processing strings compared to humans. To address this limitation, we provided a detailed analysis to uncover the underlying reasons for LLMs' struggle with string processing tasks. Building on these insights, we proposed an effective solution that enhanced LLMs' performance via fine-tuning. Our fine-tuned models increased average test accuracy of three datasets by at least 38.80%. Interesting future work includes 1) training more capable LLMs for string processing; 2) refining our datasets; and 3) further analyzing the underlying limitations of LLMs in string processing.

# 8 ACKNOWLEDGMENTS

We thank the anonymous reviewers for their very constructive comments. This work was supported by the Microsoft Accelerating Foundation Models Research program, the Commonwealth Cyber Initiative (CCI), and NSF under grants No. 2112562, 1937787, and 2125977. We also acknowledge the credits provided by Microsoft Azure.

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

# A APPENDIX

| LLMs | #Parameters | Model Provider |
|---|---|---|
| GPT-4-Turbo | Unknown | OpenAI |
| GPT-4o | Unknown | OpenAI |
| GPT-3.5 | Unknown | OpenAI |
| DeepSeek-Coder | 16B | DeepSeek |
| DeepSeek-Chat | 16B | DeepSeek |
| Gemma-2-9b-it | 9B | Google |
| Mistral-7B-v0.3 | 7B | Mistral AI |
| Mathstral-7B-v0.1 | 7B | Mistral AI |
| Codestral-22B-v0.1 | 22B | Mistral AI |
| Llama-3.1-8B-Instruct | 8B | Meta |

Table 10: Number of parameters and model providers of LLMs used in our experiments.

| Hash Function | Length |
|---|---|
| MD5 | 32 |
| SHA-1 | 40 |
| SHA-256 | 64 |
| BLAKE2b | 128 |
| SHA3-224 | 56 |
| SHAKE-128 | 32 |
| BLAKE2s | 64 |
| SHA3-512 | 128 |
| SHAKE-256 | 64 |
| SHA-384 | 96 |

Table 11: Hash functions used in our data construction and their output lengths (number of alphanumeric characters).

| Code | Variables | Templates |
|---|---|---|
| `ans = a + b` | a,b | Concat string {a} and {b}. |
| `ans = a * b` | a,b | Repeat string {a} for {b} times. |
| `ans = a[:y]` | a,y | Retrieve the first {y} characters from string {a}. |
| `ans = a[::-1]` | a | Get the reverse of the string {a}. |
| `ans = len(a)` | a | Determine the number of characters in the string {a}. |
| `ans = x in y` | x,y | Does string {y} contain substring {x}? |
| `ans = a.count(x)` | a,x | Check the occurrence of {x} within string {a}. |
| `ans = a.strip(x)` | a,x | Remove characters in {x} from both ends of string {a}. |
| `ans = a.startswith(x)` | a,x | Identify if string {a} starts with {x}. |
| `ans = a.endswith(x)` | a,x | Determine if string {a} ends with substring {x}. |

Table 12: Examples of atomic tasks.

,

Table 13: Accuracy for string processing capability of different LLMs, where each character of strings is assigned a single token ID.

| LLM | Method | Multilingual | Hash | Random String | AVG |
|---|---|---|---|---|---|
| Gemma-2-9b | Raw Inst. | 18.46 | 19.69 | 12.34 | 16.83 |
| | CoT | 20.55 | 23.59 | 16.23 | 20.12 |
| | PoT | 50.86 | 54.35 | 16.95 | 40.72 |
| Mistral-7B-v0.3 | Raw Inst. | 4.37 | 4.68 | 2.82 | 3.96 |
| | CoT | 10.19 | 10.98 | 5.22 | 8.80 |
| | PoT | 29.18 | 30.63 | 13.31 | 24.37 |
| Llama-3.1-8B | Raw Inst. | 9.62 | 17.09 | 8.11 | 11.61 |
| | CoT | 16.76 | 22.32 | 18.14 | 19.07 |
| | PoT | 38.20 | 39.06 | 17.74 | 31.67 |

**Prompt**

Use Chain-of-Thought reasoning, solve the question below step-by-step:

[Question] Retrieve string "`B2F8DbCcFfA01eC8B6E891C317C8D26a523eC0A45117E9Da0 bB2817f95A0A799`" with its characters in reverse order.

**Model Output**

To solve the problem of reversing the string `B2F8DbCcFfA01eC8B6E891C317C8D26a523eC0A45117E9Da0bB2817f95A0A799` using Chain-of-Thought reasoning, let's break down the steps:

1. **Understand the Problem:** We need to reverse the string, which means that the first character of the original string should be the last character in the reversed string, the second character should be the second last, and so on.

2. **Input String:**

`B2F8DbCcFfA01eC8B6E891C317C8D26a523eC0A45117E9Da0bB2817f95A0A799`

3. **Reversing the String:** To reverse a string, we take the last character and move towards the first character, appending each character in reverse order.

Let's break down the reversal step-by-step:

- Last character: `9`
- Second last character: `9`
- Third last character: `7`
- Continue this process until we reach the first character: `B`.

4. **Construct the Reversed String**: By appending each character from the last to the first, we get:

`997A0A59f7812Bb0aD9E71154A0Ce325a62D8C71C198E6B8C1e10AfFccDb8F2B`

5. **Verify the Reversed String:** We can check the reversed string by reading it in reverse and verifying if it matches the original string.

**Final Answer:**

The string in reverse order is:

`997A0A59f7812Bb0aD9E71154A0Ce325a62D8C71C198E6B8C1e10AfFccDb8F2B`

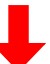

**Final Output**

`997A0A59f7812Bb0aD9E71154A0Ce325a62D8C71C198E6B8C1e10AfFccDb8F2B`

(a) CoT

Figure 2: Examples of GPT-4o solving a string processing task using different prompt engineering techniques.

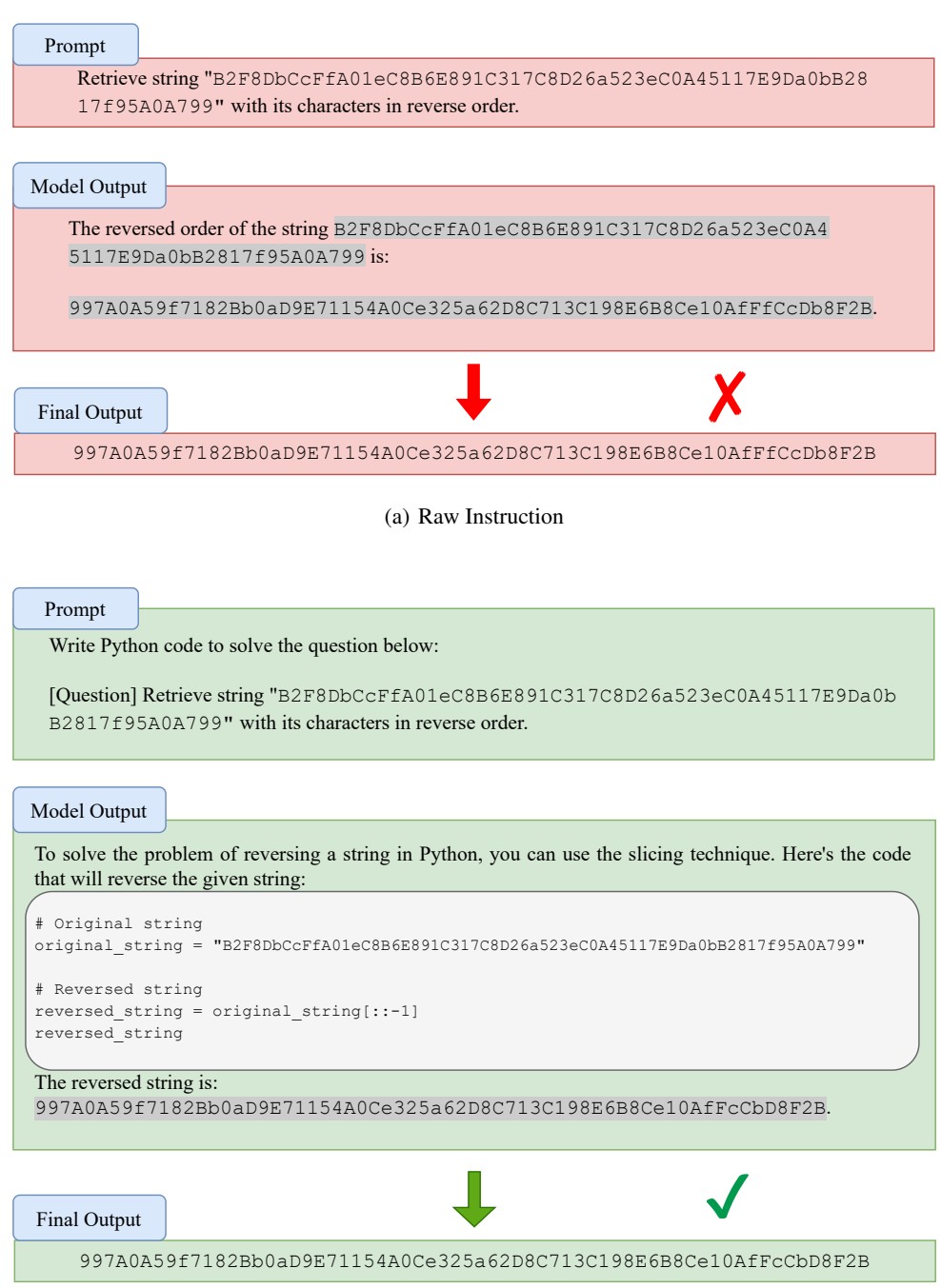

Figure 3: Examples of GPT-4o solving a string processing task using different prompt engineering techniques.

