# OpenReview forum: "StringLLM: Understanding the String Processing Capability of Large Language Models"
_ICLR.cc/2025/Conference — ICLR 2025 Poster_

### Official Review · Reviewer_rXuV · 2024-10-28

**Soundness:** 2
**Presentation:** 3
**Contribution:** 2
**Rating:** 5
**Confidence:** 4

**Summary:**

This paper proposes a set of new tasks (and a method to generate the datasets) to evaluate LLMs on string operations. String operations have been a challenge since the beginning of LLMs. In this paper, a method based on GPT-4o is proposed in order to generate large sets of evaluation data for string operations. Instead of asking the model directly, they leverage the high programming capabilities of GPT-4o to generate a program that performs the string operations, and in that way, generates the corresponding input-output pairs correctly. Regarding the evaluation, they evaluate a battery of LLMs ranging from GPT-4o to open-source alternatives like Mistral and Llama. The result suggests, as previously expected, that the LLMs struggle with character-level operations, particularly for random strings and hashes.

**Strengths:**

* The paper presents a clear experimental setup with problem definition, automatic dataset generation, and model evaluation.
* The paper is well-written and easy to follow.

**Weaknesses:**

* The major weakness of this paper comes from the motivation: it is well known that LLMs struggle with character-level operations, because, as they mention, the tokenization process removes the information about the characters. One may ask why you would ask an LLM to perform such operations, and I think it is not properly reflected in the introduction of the paper. From another perspective, they made use of an LLM to generate the evaluation dataset, which means that an LLM is capable of performing string operations, even though not directly (executing the code the LLM themselves generate). So, would not be a solution for the problem you present to provide an LLM the ability to use a Python interpreter or a predefined set of functions? I think the paper will be more strong if these problems are addressed.

**Questions:**

* **Q1**: Why is the human baseline so low when it comes to string operations in the Multilingual setting?

---

> ### Author Response · Authors · 2024-11-18
> **Response to Reviewer rXuV**
>
> We really appreciate your constructive comments and insightful suggestions. Please kindly find our response below.
>
> > Weakness 1: The major weakness of this paper comes from the motivation: it is well known that LLMs struggle with character-level operations, because, as they mention, the tokenization process removes the information about the characters. One may ask why you would ask an LLM to perform such operations, and I think it is not properly reflected in the introduction of the paper.
>
> **Response:** Thanks for your comment. When examining the Chain of Thought examples in the OpenAI o1 documentation [1], particularly the cipher and crossword examples, we observe that its reasoning chain involves multiple instances of string processing. String processing can thereby enhance the reasoning capabilities of LLMs such as o1, as these examples highlight that string processing is a crucial step in LLM reasoning. We have revised the paper accordingly. We highlight the changes in red color in the revised paper.
>
> > Weakness 2: From another perspective, they made use of an LLM to generate the evaluation dataset, which means that an LLM is capable of performing string operations, even though not directly (executing the code the LLM themselves generate).
>
> **Response:** Thanks for your comment. As stated in our initially submitted paper (now in lines 148-156, 180-183, and 190-202 of our revised paper), when constructing dataset, we rely minimally on the LLMs’ capability of performing string operations, even if we execute the code the LLMs themselves generate. Instead, we make use of its coding abilities, including code generation, understanding, and summarization.
>
> > Weakness 3: So, would not be a solution for the problem you present to provide an LLM the ability to use a Python interpreter or a predefined set of functions? I think the paper will be more strong if these problems are addressed.
>
> **Response:** Thanks for your comment. Our research focuses on enabling LLMs to solve string processing tasks independently, without relying on external functions or tools. For instance, while mathematical problems can be solved using tool-integrated approaches such as ToRA [2], there remains significant interest in studying mathematical reasoning without external tools, as seen in research efforts like o1.
>
> Besides, in our evaluation, we have already used PoT to prompt LLMs to process strings, and we execute the code the LLMs themselves generate. Therefore, a Python interpreter may not be necessary. Results show that PoT improves performance compared to CoT, but LLMs with PoT still have poor performance. This suggests that PoT might not be a good solution, alternative solutions such as fine-tuning are needed.
>
>
> > Question 1: Why is the human baseline so low when it comes to string operations in the Multilingual setting?
>
> **Response:** Thank you for your question. We have briefly explained this finding in our initially submitted paper (now in lines 320 and 321 of our revised paper): "Humans struggle to process strings manually, as they cannot identify and segment multilingual characters".
>
> To be more detailed, for example, in the case of the Arabic language, for humans whose mother tongue is not this language, it can be challenging for them to process such strings due to the difficulty in identifying every character in the string. Thus, it may be difficult for them to determine the textual form of strings and accurately process them. We have added this detailed explanation to our paper. We highlight the changes in red color in the revised paper.
>
> Thank you again for your review, and please feel free to share any additional comments or feedback.
>
> ### **References**
>
> [1] OpenAI. Learning to reason with llms, 2024. https://openai.com/index/learning-to-reason-with-llms.
>
> [2] Gou et al. Tora: A tool-integrated reasoning agent for mathematical problem solving. arXiv preprint arXiv:2309.17452, 2023

---

> ### Author Response · Authors · 2024-11-23
> **We are keen to discuss further with you**
>
> Dear Reviewer rXuV,
>
> Thank you once again for your valuable time and the constructive feedback you have provided. We sincerely appreciate your insights and remain eager to engage in further discussions to address any remaining concerns.
>
> As the deadline for the discussion period is approaching, we would be grateful if you could kindly let us know if there are any additional questions or points requiring clarification.
>
> Thank you for your continued attention to our work.
>
> Best regards,
>
> The Authors

---

> > ### Comment · Reviewer_rXuV · 2024-11-23
> >
> > Thank you for your response. However, I still have concerns regarding the motivation of the paper. Do you believe that teaching skills that can be programmed to LLMs will be the future of LLMs? Machine learning is interesting because it allows us to train estimators for tasks that we cannot program directly. However, what are the advantages of asking a model to solve one task that can be programmed? That is, why would you choose the option that can fail sometimes and is not consistent over different runs...
> >
> > I will increase my score a bit because even though that is not the future (in my point of view) it can be used to evaluate the programming or tool using skills of LLMs.
> >
> > **Regarding other comments**
> > > To be more detailed, for example, in the case of the Arabic language, for humans whose mother tongue is not this language, it can be challenging for them to process such strings due to the difficulty in identifying every character in the string.
> >
> > I see.
> >
> > **Additional questions**
> > * In Table 7, where did you find the number of parameters for GPT-4-Turbo, GPT-4o, and GPT-3.5? Can you give a reference?

---

> ### Author Response · Authors · 2024-11-23
> **Response to Reviewer rXuV**
>
> # **Thank you for your comment and for raising our score!**
>
> > Comment: However, I still have concerns regarding the motivation of the paper. Do you believe that teaching skills that can be programmed to LLMs will be the future of LLMs? Machine learning is interesting because it allows us to train estimators for tasks that we cannot program directly. However, what are the advantages of asking a model to solve one task that can be programmed? That is, why would you choose the option that can fail sometimes and is not consistent over different runs...
>
> **Response:** Thanks for your insightful comment. We agree that solving tasks that we cannot program directly is an interesting future direction. In fact, a real-world task often consists of both programmable and non-programmable subtasks, where programmable subtasks provide context for non-programmable ones and influence their resolution. Consequently, poorly addressing programmable subtasks may negatively impact the resolution of non-programmable subtasks. Therefore, we also believe that accurately solving programmable subtasks will still be important in the future.
>
> Again take the Chain of Thought examples in the OpenAI o1 documentation [1]. The cyber task illustrates how a task can consist of both programmable subtasks (e.g., string processing) and non-programmable subtasks (e.g., reasoning). In this case, string processing provides the context for reasoning and can thereby influence how OpenAI o1 performs the reasoning process.
>
> Besides, we also believe using LLMs as assistants to humans, performing tasks that humans can do such as reasoning and programming, thereby reducing the cognitive and physical effort required of individuals, would be another interesting direction.
>
>
> > Additional question: In Table 7, where did you find the number of parameters for GPT-4-Turbo, GPT-4o, and GPT-3.5? Can you give a reference?
>
> **Response**: Some blogs [2, 3] report the number of parameters of the GPT family. However, we believe these are just estimations and not official since the OpenAI does not disclose the exact number of parameters of the GPT family.
>
> Thank you again for your review, and please feel free to share any additional comments or feedback.
>
> ### References
>
> [1] OpenAI. Learning to reason with llms, 2024. https://openai.com/index/learning-to-reason-with-llms.
>
> [2] Josh Howarth. Number of Parameters in GPT-4 (Latest Data). https://explodingtopics.com/blog/gpt-parameters
>
> [3] Matthias Bastian. GPT-4 has more than a trillion parameters. https://the-decoder.com/gpt-4-has-a-trillion-parameters/

---

> > ### Comment · Reviewer_rXuV · 2024-11-24
> >
> > Adding that information (the parameters of GPT-4) is imprudent, but, in any case, it should not be written in a paper without at least a reference to the source and a disclaimer that is an estimation. I suggest removing any information you are not 100% sure about.

---

> > > ### Author Response · Authors · 2024-11-24
> > > **Response to Reviewer rXuV**
> > >
> > > Thanks for your suggestion. We have revised our paper accordingly.

---

### Official Review · Reviewer_BYDo · 2024-11-01

**Soundness:** 2
**Presentation:** 3
**Contribution:** 2
**Rating:** 6
**Confidence:** 4

**Summary:**

This paper is concerned with the string processing abilities of large language models, and the paper introduces a novel benchmark for quantifying string processing abilities. The benchmark is used to compare a number of existing models, and it is shown that finetuning can improve the performance of models on the benchmark.

**Strengths:**

The paper present a new benchmark for quantifying string processing abilities of large language models. This is a novel contribution that may be helpful to rank and select models for certain types of practical applications.

**Weaknesses:**

The claim regarding the multilinguality of the dataset is not convincing, since it only pertains to the strings and not the questions in the dataset.

There are some redundant material in the paper, e.g. the section on how LLMs analyze input text via Transformers can be removed since it is common knowledge in the community, and so is the cosine similarity.

I do not agree with the intuition that the token embeddings should contain information about token lengths, and as such the results in section 5.2 are not very surprising. Even so, I do not agree with the conclusion that this explains why models fail to perform string processing tasks, since my intuition is that this is more or less completely dependent on tokenization.

The paper shows that results can be improved by finetuning, but there are no comparison with alternative approaches that operate on the tokenizer level rather than updating the whole model.

**Questions:**

How much of the current weaknesses of models to handle string processing tasks is related to relatively simple facts and practices related to tokenization? For example, how would if affect the results if all strings in the test data were replaced with string that featured white space between all characters?

Regarding multilinguality, even if the data contains strings from different languages, all questions are still in English. Would it not make sense to also include multilingual instructions in the data if the goal is to measure multilingual capacity?

The experiments in section 6 includes the use of some general-purpose datasets. What would happen if these were not included? The conclusion in this section states that "our finetuning not only enhances LLM's string processing ability but also preserves their foundational capabilities". But one suspects that this is only due to having included the general-purpose datasets in the finetuning?

---

> ### Author Response · Authors · 2024-11-18
> **Response to Reviewer BYDo [Weaknesses]**
>
> We really appreciate your constructive comments and insightful suggestions. Please kindly find our response below.
> > Weakness 1: The claim regarding the multilinguality of the dataset is not convincing, since it only pertains to the strings and not the questions in the dataset.
>
> **Response:** Thanks for your valuable insights. However, the focus of our study is on analyzing the string processing capabilities of LLMs across different language distributions, not on the ability to follow instructions in multiple languages. By using English for all instructions, we maintain a consistent format that isolates and highlights the string processing aspect, ensuring that the evaluation is strictly on assessing the LLMs’ capability of processing multilingual string, rather than instruction comprehension in various languages.
>
> > Weakness 2: There are some redundant material in the paper, e.g. the section on how LLMs analyze input text via Transformers can be removed since it is common knowledge in the community, and so is the cosine similarity.
>
> **Response:** Thanks for your insightful comment. We have revised our paper accordingly.
>
> > Weakness 3: I do not agree with the intuition that the token embeddings should contain information about token lengths, and as such the results in section 5.2 are not very surprising. Even so, I do not agree with the conclusion that this explains why models fail to perform string processing tasks, since my intuition is that this is more or less completely dependent on tokenization.
>
> **Response:** We appreciate the reviewer’s perspective. We agree that string processing is more or less dependent on tokenization. Given the limitation of Section 5.2, we decided to revise our paper, **removing Section 5.2** and **replacing this section with the alternative analysis** described in our response to **Question 1**.  We highlight the changes in red color in the revised paper.
>
> > Weakness 4: The paper shows that results can be improved by finetuning, but there are no comparison with alternative approaches that operate on the tokenizer level rather than updating the whole model.
>
> **Response:** Thank you for your insightful comment. Below are **two brief experiments** with alternative approaches that operate on the tokenizer level.
>
> For **the first analysis**, the Table below shows LLMs' performance where the characters in the string are encoded separately, meaning each character is encoded by its token ID in the vocabulary, rather than being processed as part of a complete word or subword. We have added these new results in Table 13 of the Appendix of our revised paper.
>
> | LLM          | Method   | Multilingual | Hash  | Random String | AVG |
> |--------------|:----------|:--------------:|:-------:|:---------------:|:--------:|
> | Gemma-2-9b   | Raw Inst. | 18.46       | 19.69 | 12.34         |   16.83   |
> |              | CoT      | 20.55        | 23.59 | 16.23         | 20.12 |
> |              | PoT      | 50.86        | 54.35 | 16.95         |  40.72   |
> | Mistral-7B-v0.3 | Raw Inst. | 4.37    |  4.68   | 2.82         |  3.96   |
> |              | CoT      | 10.19        | 10.98 |  5.22          | 8.80 |
> |              | PoT      | 29.18        | 30.63 | 13.31         | 24.37 |
> | Llama-3.1-8B | Raw Inst. | 9.62       | 17.09 | 8.11         | 11.61 |
> |              | CoT      | 16.76        | 22.32 | 18.14         | 19.07 |
> |              | PoT      | 38.20        | 39.06 | 17.74         | 31.67 |
>
> Surprisingly, the performance of LLMs degrades compared to Table 3 in the paper. One possible explanation is that this essentially employs a different tokenizer, as it encodes the same input text in a different way. The performance drops since LLMs are trained on the original tokenizer. Retraining an LLM with a new tokenizer would be another alternative approach. However, it requires significant computational resources, which are not available to us. We have acknowledged this limitation in our revised paper and emphasized it as a potential future direction. We highlight the changes in red color in the revised paper.
>
> ## **For another analysis of tokenization, please refer to our response to Question 1 below.**
>
> ---
>
> Thank you again for your review, and please feel free to share any additional comments or feedback.

---

> ### Author Response · Authors · 2024-11-18
> **Response to Reviewer BYDo [Questions]**
>
> > Question 1: How much of the current weaknesses of models to handle string processing tasks is related to relatively simple facts and practices related to tokenization? For example, how would if affect the results if all strings in the test data were replaced with string that featured white space between all characters?
>
> **Response:** Thanks for the question. We conducted the suggested experiment and the results are shown as follows. The Table below shows the string processing capability of different LLMs, where white spaces are inserted between all characters of strings. This insertion of white spaces aims to segment every character in strings, avoiding the use of word-level or subword-level tokens, thereby introducing potential character-level information to LLMs.
>
> | LLM         | Method   | Multilingual | Hash  | Random String |AVG|
> |:-------------|:----------|:--------------:|:-------:|:---------------:|:----:|
> | Gemma-2-9b  | Raw Inst.| 20.48        | 21.23 | 14.31         | 18.67 |
> |             | CoT      | 24.22        | 25.16 | 20.07         |  23.15 |
> |             | PoT      | 55.69        | 52.73 | 18.98         | 42.47 |
> | Mistral-7B-v0.3 | Raw Inst. | 4.97     | 3.99  | 2.79          | 3.92 |
> |             | CoT      | 12.45        | 10.53 | 7.34          | 10.11 |
> |             | PoT      | 32.28        | 32.67 | 15.32         | 26.76 |
> | Llama-3.1-8B| Raw Inst.| 12.83        | 16.56 | 9.76          | 13.05 |
> |             | CoT      | 20.64        | 23.21 | 19.20         | 21.02 |
> |             | PoT      | 42.01        | 43.30 | 24.74         | 36.68 |
>
> Compared to Table 3 in our paper, the performance of LLMs is improved, indicating that the word-level or subword-level tokens make LLMs lack character-level understanding. In contrast, augmenting tokenizer with finer-grained character-level segmentation could be a promising direction for improving LLM performance. **However, the performance of LLMs just showed slightly improvement, indicating that the current weaknesses of models to handle string processing tasks are probably not solely related to relatively simple facts and practices related to tokenization. Hence, further solutions such as finetuning are needed.**
>
> **We have added this new analysis in Section 5.2 of our revised paper. We highlight the changes in red color in the revised paper.**
>
> > Question 2: Regarding multilinguality, even if the data contains strings from different languages, all questions are still in English. Would it not make sense to also include multilingual instructions in the data if the goal is to measure multilingual capacity?
>
> **Response:** Please refer to our response on Weakness 1.
>
> > Question 3: The experiments in section 6 includes the use of some general-purpose datasets. What would happen if these were not included? The conclusion in this section states that "our finetuning not only enhances LLM's string processing ability but also preserves their foundational capabilities". But one suspects that this is only due to having included the general-purpose datasets in the fine-tuning?
>
> **Response:** Thanks for your question. Coding LLMs, such as CodeLlama [1] and DeepSeek-Coder [2], typically include general datasets in their training phases. The inclusion of such datasets is a common practice to ensure the models have a certain level of foundational capabilities. Hence, excluding general datasets would compromise the foundational capabilities of the models. We revised our paper to clarify this point. We highlight the changes in red color in the revised paper.
>
> The Table below shows the performance where LLMs are finetuned without general-purpose datasets. We have added these new results in Table 9 of our revised paper.
>
> | LLM    |       | MMLU | HellaSwag | ARC | Avg. Change |
> | :-| :- | :-: | :-: | :-: | :-: |
> | Gemma-2-9b | Before       |   71.88 | 80.08 | 64.76   |    |
> |    | After        |  71.41  |    79.76   |  61.09   |      -1.49      |
> | Mistral-7B-v0.3 |  Before       | 59.74 |  82.90 | 58.62  |        |
> |     | After       |  57.72  |   82.52    | 56.23  |       -1.60     |
> | Llama-3.1-8B  |   Before     |  67.94 | 79.33 | 55.03  |       |
> |    | After       |  65.65  |   80.60    |  54.68  |   - 0.46         |
>
> Compared to Table 7 in our paper, excluding general datasets would compromise the foundational capabilities of the models, which is as expected.
>
> Thank you again for your review, and please feel free to share any additional comments or feedback.
>
> ### **References**
>
> [1] Roziere et al. Code llama: Open foundation models for code. arXiv preprint arXiv:2308.12950, 2023.
>
> [2] Zhu et al. DeepSeek-Coder-V2: Breaking the barrier of closed-source models in code intelligence. arXiv preprint arXiv:2406.11931, 2024.

---

> ### Author Response · Authors · 2024-11-23
> **We are keen to discuss further with you**
>
> Dear Reviewer BYDo,
>
> Thank you once again for your valuable time and the constructive feedback you have provided. We sincerely appreciate your insights and remain eager to engage in further discussions to address any remaining concerns.
>
> As the deadline for the discussion period is approaching, we would be grateful if you could kindly let us know if there are any additional questions or points requiring clarification.
>
> Thank you for your continued attention to our work.
>
> Best regards,
>
> The Authors

---

> > ### Comment · Reviewer_BYDo · 2024-11-26
> >
> > Thanks you for your detailed responses and additional experiments. I am satisfied with the answers, and am willing to raise my score on point based on your answers.

---

> > > ### Author Response · Authors · 2024-11-26
> > > **A gentle reminder to raise rating score**
> > >
> > > Dear Reviewer,
> > >
> > > Thank you once again for your review and constructive feedback. Your pre-rebuttal rating was 5. We’re delighted to know that you found our rebuttal and new experiments satisfactory, and we appreciate your kind promise to raise your rating. As a gentle reminder, we kindly ask that you update your rating at your convenience to ensure your post-rebuttal evaluation is accurately reflected in the system. Thank you again for your time and support!
> > >
> > > Best,
> > > Authors

---

> ### Author Response · Authors · 2024-11-27
> **A gentle reminder to raise rating score**
>
> Dear Reviewer,
>
> Thank you once again for your constructive feedback! Your pre-rebuttal rating was 5. We greatly appreciate your kind promise to raise your rating.  As a gentle reminder, we kindly ask that you update your rating at your convenience to ensure your post-rebuttal evaluation is accurately reflected in the system. Thank you again for your time and support!!
>
>
> Best,
>
> Authors

---

> ### Author Response · Authors · 2024-12-02
> **A gentle reminder to raise rating score**
>
> Dear Reviewer BYDo,
>
> Thank you once again for your constructive feedback. Your pre-rebuttal rating was 5. We greatly appreciate your kind promise to raise your rating. However, we have noticed that the rating has not yet been updated. As the deadline of rebuttal is approaching, we kindly request that you update your rating at your earliest convenience to ensure your post-rebuttal evaluation is accurately reflected in the system. Thank you for your time and continued support.
>
> Best,
>
> Authors

---

### Official Review · Reviewer_iEx9 · 2024-11-03

**Soundness:** 2
**Presentation:** 2
**Contribution:** 3
**Rating:** 6
**Confidence:** 4

**Summary:**

1. The authors presented a methodology to construct a large scale string manipulation dataset, consisting of sets of (question, code, answer) tuples. This is done by first hand crafting a set of code templates, then generate natural language instruction templates describing the code template using GPT, and finally getting the final answers by filling in variables with different types of strings and executing those code.
2. They presented an analysis attempting to explain why LLMs struggle with string processing tasks.
3. They demonstrated that fine-tuning improves LLM performance on this task.

**Strengths:**

1. I appreciate the effort to address this problem. This seems like a useful benchmark for improving LLMs code generation ability for string manipulation
2. They presented a clever way of leveraging closed source LLMs for constructing a dataset, and made a good and thorough effort at demonstrating that it poses a challenge to current LLMs.
3. They showed PoT fine-tuning helps without hurting performance on other NLU tasks.

**Weaknesses:**

The authors only reported performance on MMLU/ARC/HellaSwag and show that it only degrades a little. I think since this is finetuned on code gen specifically for string manipulation, this should also be benchmarked on code generation benchmarks such as HumanEval, MBPP, CoNaLa, or the newer BigCodeBench/EvalPlus to make sure it doesn't hinder performance in those areas.

**Questions:**

Overall, I think the paper made some good contributions, but there are some missing experiments. I am happy to revise my evaluation based on author response.


Below is just a small nitpick, though to me it doesn't really undermine the contribution of the paper, since they mainly used PoT to get around the problem.

In section 5.2. authors claimed that since there is no observable diagonal pattern in the embedding similarity matrix in figure 2b, it suggests that token embedding lacks information on character-level token length. I fail to see how this is the case.

Consider a toy example with just 4 tokens. Let's design the embedding s.t. it is 2d and the first dimension always encode the character length. Let's say these four vectors are (grouped by character length) E1 = [[1,1],[1,-1]] and E2=[[2,2],[2,-2]]. By definition we would get Sim(E1,E1) = 0, Sim(E2,E2)=0, but Sim(E1,E2)=0.5, where there is NO diagnoal pattern, but by construction it captures character level token length.

Maybe the authors can consider proposing an alternative analysis to strengthen their claim that tokens lack character level information.

---

> ### Author Response · Authors · 2024-11-18
> **Response to Reviewer iEx9**
>
> We really appreciate your constructive comments and insightful suggestions.
>
>  > Weakness 1: The authors only reported performance on MMLU/ARC/HellaSwag and show that it only degrades a little. I think since this is finetuned on code gen specifically for string manipulation, this should also be benchmarked on code generation benchmarks such as HumanEval, MBPP, CoNaLa, or the newer BigCodeBench/EvalPlus to make sure it doesn't hinder performance in those areas.
>
> **Response:** We thank the reviewer for your constructive comments. We have added an experiment according to your suggestion:
> | LLM    |        | HumanEval | HumanEval+ | MBPP | MBPP+ | Avg. Change |
> | :-| :- | :-: | :-: | :-: | :-: | :-: |
> | Gemma-2-9b | Before       | 67.7      | 59.1       | 73.3 | 63.0  |       |
> |    | After        | 66.5      | 59.1       | 72.8 | 62.2  |   -0.78           |
> | Mistral-7B-v0.3 |  Before       | 36.0      | 31.1       | 50.3 | 42.1  |        |
> |     | After        | 34.8      | 30.5       | 49.2 | 42.3  |    -0.68         |
> | Llama-3.1-8B  |   Before       | 64.5      | 55.5       | 68.0 | 55.6  |       |
> |    | After        | 63.4      | 54.3       | 67.5 | 54.2  |     -1.05     |
>
>
> As shown in the results, it's evident that the code generation performance of LLMs also only shows a slight decline (no more than 1.05% on average) before and after fine-tuning. However, as stated in the paper, our fine-tuning method is quite naive, since it is not our primary focus. AI companies with advanced model training pipelines and substantial computational resources can better integrate our dataset with general and code-specific data, enabling them to maintain both the general and coding capabilities of LLMs. We have added these new results in Table 10 in the Appendix of our revised paper.
>
>
> > Question 1: Below is just a small nitpick, though to me it doesn't really undermine the contribution of the paper, since they mainly used PoT to get around the problem.
> >
> > In section 5.2. authors claimed that since there is no observable diagonal pattern in the embedding similarity matrix in figure 2b, it suggests that token embedding lacks information on character-level token length. I fail to see how this is the case.
> >
> > Consider a toy example with just 4 tokens. Let's design the embedding s.t. it is 2d and the first dimension always encode the character length. Let's say these four vectors are (grouped by character length) E1 = [[1,1],[1,-1]] and E2=[[2,2],[2,-2]]. By definition we would get Sim(E1,E1) = 0, Sim(E2,E2)=0, but Sim(E1,E2)=0.5, where there is NO diagnoal pattern, but by construction it captures character level token length.
> >
> > Maybe the authors can consider proposing an alternative analysis to strengthen their claim that tokens lack character level information.
>
> **Response:** Thanks for the insightful comment and example. We acknowledge this limitation and appreciate the suggestion for strengthening our analysis. Our analysis shows that the whole token embedding does not encode length information. However, it does not exclude the possibility that a subset of embedding dimensions encodes token length information. Nevertheless, identifying such a subset may be computationally intractable. We have revised our paper to clarify this point and have added the corresponding statement to ensure greater clarity. We highlight the changes in red color in the revised paper.
>
> The following is a brief alternative analysis. The Table below shows the string processing capability of different LLMs, where white spaces are inserted between all characters of strings. This insertion of white spaces aims to segment every character in strings, avoiding the use of word-level or subword-level tokens, thereby introducing potential character-level information to LLMs.
>
> | LLM         | Method   | Multilingual | Hash  | Random String |AVG|
> |:-|:-|:-:|:-:|:-:|:-:|
> | Gemma-2-9b  | Raw Inst.| 20.48        | 21.23 | 14.31         | 18.67 |
> |             | CoT      | 24.22        | 25.16 | 20.07         |  23.15 |
> |             | PoT      | 55.69        | 52.73 | 18.98         | 42.47 |
> | Mistral-7B-v0.3 | Raw Inst. | 4.97     | 3.99  | 2.79          | 3.92 |
> |             | CoT      | 12.45        | 10.53 | 7.34          | 10.11 |
> |         | PoT      | 32.28        | 32.67 | 15.32         | 26.76 |
> | Llama-3.1-8B| Raw Inst.| 12.83| 16.56 | 9.76| 13.05 |
> |   | CoT | 20.64 | 23.21 | 19.20| 21.02 |
> |    | PoT | 42.01 | 43.30 | 24.74| 36.68 |
>
> Compared to Table 3 in our paper, the performance of LLMs is improved, indicating that the word-level or subword-level tokens make LLMs lack character-level understanding. In contrast, augmenting tokenizer with finer-grained character-level segmentation could be a promising direction for improving LLM performance. We have added these new results in Table 11 of the Appendix of our revised paper.
>
> Thanks again for your review, and please feel free to share any additional comments or feedback.

---

> > ### Comment · Reviewer_iEx9 · 2024-11-25
> >
> > I appreciate the authors taking the time to run the suggested experiments.
> > I am willing to bump up my score since I think this is a somewhat useful benchmark.
> > Furthermore, despite the finetuning approach, as the authors have also acknowledged, being quite naive, I think the results they reported from experiments suggested by [reviewer b4nW](https://openreview.net/forum?id=kTXChtaaNO&noteId=MRaRNGQ435) and [reviewer BYDo](https://openreview.net/forum?id=kTXChtaaNO&noteId=Oc2iomNXQ4) are sufficiently interesting additions to the naive finetuning experiments.
> >
> > However, I still believe the revisions to section 5.2 is flawed. The authors are claiming that, if (p=LLM token embeddings encode character information), then the (q=cosine similarity of embeddings between different length groups should have a diagonal pattern). Your experiment shows lack of such pattern, and therefore you claim by the contrapositive (~q => ~p) that LLM's token embedding does not encode character information. The proposition (p=>q) is simply taken for granted, hence I raised the counter example. Even with the modified antecedent ("p") in your revision, you still have not established the causality of p=>q.
> > Why should (whole) token embeddings encoding character info necessitate high cosine similarity in same character length token embeddings? I think maybe figure 2b and the corresponding text should be removed and make room for other interesting analyses you conducted in your final revision.

---

> ### Author Response · Authors · 2024-11-23
> **We are keen to discuss further with you**
>
> Dear Reviewer iEx9,
>
> Thank you once again for your valuable time and the constructive feedback you have provided. We sincerely appreciate your insights and remain eager to engage in further discussions to address any remaining concerns.
>
> As the **deadline** for the discussion period is approaching, we would be grateful if you could kindly let us know if there are any additional questions or points requiring clarification.
>
> Thank you for your continued attention to our work.
>
> Best regards,
>
> The Authors

---

> ### Author Response · Authors · 2024-11-25
> **Response to Reviewer iEx9**
>
> # **Thank you for your insightful comment and for raising our score!**
>
> > Comment: However, I still believe the revisions to section 5.2 is flawed. The authors are claiming that, if (p=LLM token embeddings encode character information), then the (q=cosine similarity of embeddings between different length groups should have a diagonal pattern). Your experiment shows lack of such pattern, and therefore you claim by the contrapositive (~q => ~p) that LLM's token embedding does not encode character information. The proposition (p=>q) is simply taken for granted, hence I raised the counter example. Even with the modified antecedent ("p") in your revision, you still have not established the causality of p=>q. Why should (whole) token embeddings encoding character info necessitate high cosine similarity in same character length token embeddings? I think maybe figure 2b and the corresponding text should be removed and make room for other interesting analyses you conducted in your final revision.
>
> **Response:** Thank you for your insightful feedback. Upon reflection and further discussion, we acknowledge the shortcomings of Section 5.2. We appreciate your detailed critique regarding the causality of p=>q and agree that this relationship was not sufficiently justified in our analysis. As suggested, we have removed Section 5.2, including Figure 2b and its corresponding text, and moved the suggested alternative analysis to the main body in the final revision.

---

### Official Review · Reviewer_b4nW · 2024-11-03

**Soundness:** 4
**Presentation:** 3
**Contribution:** 4
**Rating:** 8
**Confidence:** 4

**Summary:**

First comprehensive study of string processing capabilities of LLMs. Aside from just evaluating, authors also proposed specific solutions (via fine-tuning) to improve their capabilities. Critically, while this fine-tuning does improve their string processing capabilities significantly, it does not majorly harm their general-purpose aibilities.

**Strengths:**

This work handles a very specific problem, and it does so well: well-principled, reasonable evaluation, and useful solution. The analysis of the errors is also interesting.

String processing capabilities have an indirect effect on all LLMs tasks, and a direct effect on some of them, so this work is clearly relevant.

Additionally, given the fact that authors are willing to publish data and code, this works seems reproducible.

**Weaknesses:**

The main source of LLMs struggle with string processing is tokenization (this is not news). Given that, I'd have liked to see more studies on tokenization. Perhaps pretraining small LMs on different tokenization strategies.

The presentation of this paper is good enough, but it doesn't have the excellence on this aspect one would like to see on an ICLR paper. For example, Figure 2 is very relevant to the message of the paper, yet it's badly formatted/presented.

**Questions:**

Please rethink Figure 2. Same data, same basic idea, but better presentation. For example, do we really need 2 digits for the decimals on b)? Do we even need numbers there at all, given the colors?

---

> ### Author Response · Authors · 2024-11-18
> **Response to Reviewer b4nW**
>
> We really appreciate your constructive comments and insightful suggestions. Please kindly find our response below.
>
> > Weakness 1: The main source of LLMs struggle with string processing is tokenization (this is not news). Given that, I'd have liked to see more studies on tokenization. Perhaps pretraining small LMs on different tokenization strategies.
>
> **Response:** We sincerely appreciate the reviewer's valuable insights. We have conducted two more studies on tokenization.
>
> The Table below shows the string processing capability of different LLMs, where white spaces are inserted between all characters of strings. This insertion of white spaces aims to segment every character in strings, avoiding the use of word-level or subword-level tokens, thereby introducing potential character-level information to LLMs.
>
> | LLM         | Method   | Multilingual | Hash  | Random String |AVG|
> |:-|:-|:-:|:-:|:-:|:-:|
> | Gemma-2-9b  | Raw Inst.| 20.48        | 21.23 | 14.31         | 18.67 |
> |             | CoT      | 24.22        | 25.16 | 20.07         |  23.15 |
> |             | PoT      | 55.69        | 52.73 | 18.98         | 42.47 |
> | Mistral-7B-v0.3 | Raw Inst. | 4.97     | 3.99  | 2.79          | 3.92 |
> |             | CoT      | 12.45        | 10.53 | 7.34          | 10.11 |
> |         | PoT      | 32.28        | 32.67 | 15.32         | 26.76 |
> | Llama-3.1-8B| Raw Inst.| 12.83| 16.56 | 9.76| 13.05 |
> |   | CoT | 20.64 | 23.21 | 19.20| 21.02 |
> |    | PoT | 42.01 | 43.30 | 24.74| 36.68 |
>
> Compared to Table 3 in our paper, the performance of LLMs is improved, indicating that the word-level or subword-level tokens make LLMs lack character-level understanding. In contrast, augmenting tokenizer with finer-grained character-level segmentation could be a promising direction for improving LLM performance. We have added these new results in Section 5.2 of our revised paper. We highlight the changes in red color in the revised paper.
>
> ---
>
> The Table below shows LLMs' performance where the characters in the string are encoded separately, meaning each character is encoded by its token ID in the vocabulary, rather than being processed as part of a complete word or subword. We have added these new results in Table 13 of the Appendix of our revised paper.
>
> | LLM          | Method   | Multilingual | Hash  | Random String | AVG |
> |--------------|:----------|:--------------:|:-------:|:---------------:|:--------:|
> | Gemma-2-9b   | Raw Inst. | 18.46       | 19.69 | 12.34         |   16.83   |
> |              | CoT      | 20.55        | 23.59 | 16.23         | 20.12 |
> |              | PoT      | 50.86        | 54.35 | 16.95         |  40.72   |
> | Mistral-7B-v0.3 | Raw Inst. | 4.37    |  4.68   | 2.82         |  3.96   |
> |              | CoT      | 10.19        | 10.98 |  5.22          | 8.80 |
> |              | PoT      | 29.18        | 30.63 | 13.31         | 24.37 |
> | Llama-3.1-8B | Raw Inst. | 9.62       | 17.09 | 8.11         | 11.61 |
> |              | CoT      | 16.76        | 22.32 | 18.14         | 19.07 |
> |              | PoT      | 38.20        | 39.06 | 17.74         | 31.67 |
>
> Surprisingly, the performance of LLMs degrades compared to Table 3 in the paper. One possible explanation is that this essentially employs a different tokenizer, as it encodes the same input text in a different way. The performance drops since LLMs are trained on the original tokenizer. Unfortunately, we are unable to pretrain small LMs with new tokenizers due to limited computational resources. But we believe this is interesting future work.
>
> > Weakness 2: The presentation of this paper is good enough, but it doesn't have the excellence on this aspect one would like to see on an ICLR paper. For example, Figure 2 is very relevant to the message of the paper, yet it's badly formatted/presented.
>
> **Response:** We sincerely appreciate the reviewer's valuable insights. We have revised the figure based on your suggestions and updated the paper accordingly.
>
> > Question 1: Please rethink Figure 2. Same data, same basic idea, but better presentation. For example, do we really need 2 digits for the decimals on b)? Do we even need numbers there at all, given the colors?
>
> **Response:** Please refer to our response of weakness 2.
>
> Thank you again for your review, and please feel free to share any additional comments or feedback.

---

> > ### Comment · Reviewer_b4nW · 2024-11-26
> > **Response to authors**
> >
> > I appreciate your updates. The results inserting whitespaces are interesting. "We are unable to pretrain small LMs with new tokenizers due to limited computational resources", understandable.
> >
> > I'll update my ratings.

---

> ### Author Response · Authors · 2024-11-23
> **We are keen to discuss further with you**
>
> Dear Reviewer b4nW,
>
> Thank you once again for your valuable time and the constructive feedback you have provided. We sincerely appreciate your insights and remain eager to engage in further discussions to address any remaining concerns.
>
> As the **deadline** for the discussion period is approaching, we would be grateful if you could kindly let us know if there are any additional questions or points requiring clarification.
>
> Thank you for your continued attention to our work.
>
> Best regards,
>
> The Authors

---

> ### Author Response · Authors · 2024-11-26
> **Thank you very much**
>
> Thanks for your insightful comments and for updating the Confidence score from 3 to 4!

---

### Meta-Review · Area_Chair_deVx · 2025-01-03

**Metareview:**

This paper introduces StringLLM, a method for generating datasets to benchmark string processing capabilities of LLMs. Using StringLLM, the authors create StringBench, a suite of string manipulation tasks. Evaluation shows that LLM performance on StringBench tasks trails humans. To address this, the paper proposes and demonstrates a fine-tuning approach that substantially improves LLM performance.

Reviewers agreed that StringLLM and StringBench would be useful to the community. One major concern was lack of study on tokenization. The authors addressed this by presenting experimental results with character-wise tokenization through the insertion of whitespace between characters. Another concern, raised by Reviewer iEx9, was the absence of evaluation on code generation benchmarks. This was addressed by the authors by including results on HumanEval, HumanEval+, MBPP, and MBPP+ in the rebuttal. Two reviewers identified issues with the analysis presented in Section 5.2, which the authors acknowledged and committed to remove it from the paper. Other concerns include claim regarding the multilinguality of the dataset unconvincing and the study not being motivated enough.

Given the paper's contribution of a useful and well-principled benchmark, reasonable evaluation methodology, commitment to releasing data and code for easy reproducibility, and the authors addressing most of the reviewer concerns, I am recommending acceptance.

**Additional Comments On Reviewer Discussion:**

See above.

---

### Decision · Program_Chairs · 2025-01-22

Accept (Poster)